# Celiac Disease Dietary Adherence on the Rural–Urban Continuum

**DOI:** 10.3390/nu15214535

**Published:** 2023-10-26

**Authors:** Amy Posterick, Candace L. Ayars

**Affiliations:** College of Graduate Health Studies, A.T. Still University, 800 W. Jefferson St., Kirksville, MO 63501, USA; cayars@atsu.edu

**Keywords:** celiac disease, gluten-free diet, dietary adherence, rural health behaviors

## Abstract

Poor adherence to a gluten-free diet for those with celiac disease is a well-established risk factor, leading to gastrointestinal symptoms, malabsorption of nutrients, and psychiatric complications. Previous studies have shown that those outside urban areas encounter unique barriers to dietary adherence and are less likely to engage in health management behaviors than those in urban regions. This study aimed to examine the relationship between gluten-free dietary adherence and individual, relationship, and community factors, including the geographic location of residence on the rural–urban continuum, for 253 adults with celiac disease living in the United States. Those with celiac disease residing in urban regions had significantly better dietary adherence than those residing in nonurban areas (*p* < 0.05). Those living in nonurban communities had, on average, poor enough adherence scores to suggest ongoing intestinal damage from gluten consumption. Geographic location, age, years since diagnosis, and annual income significantly predicted compliance with a gluten-free diet for those with celiac disease, accounting for nearly 20% of the variance. Those living outside urban areas with a lower income, younger age, and more recent diagnosis of celiac disease had the worst dietary adherence, placing them at the most risk for ongoing disease progression and complications.

## 1. Introduction

Celiac disease is a chronic autoimmune condition with no medical treatment other than strict adherence to a gluten-free diet [1]. No medication options are available for disease management, without improved dietary adherence [2]. Celiac disease is a lifelong condition without a medical cure or resolution [1,3]. Individuals do not grow out of the disease with age or find relief or remission apart from making dietary changes [1]. Adequate adherence to a gluten-free diet is critical for reducing symptoms and improving the health-related quality of life of those with celiac disease [4,5]. A lack of adherence to a gluten-free diet for someone with celiac disease will cause ongoing disease progression, including continuing damage to the small intestine, inflammation, and increased risk of disease complications [5]. Because the treatment for celiac disease is entirely behavioral, it is essential to conduct research to identify the factors that predict the dietary adherence of those with the disease, along with potential barriers to dietary compliance.

Medical researchers estimate that for individuals with celiac to avoid inflammation from the disease reaction, they must be exposed to less than 10 milligrams of gluten per day [6]. However, a meta-analysis conducted in 2018 found that the mean exposure to gluten per day for those with celiac disease was 15–40 times more than that acceptable level [7]. Further analysis found that those with moderate to severe symptoms ingested significantly more than 200 milligrams of gluten daily [7]. Additionally, researchers examining stool samples from patients with celiac disease found that 29.8% had detectable gluten immunogenic peptides in their sample, suggesting recent gluten consumption [8]. These results suggest that many individuals with celiac disease regularly consume excessive levels of gluten, causing disease symptoms and ongoing intestinal damage.

The exact rate of adherence to a gluten-free diet among celiac patients is highly variable across studies. For example, a survey of over 5000 celiac disease patients in New Zealand and Australia found that 61% of participants were adherent to a gluten-free diet [4]. Contrastingly, a study of adult celiac patients in Canada found that fewer than 50% of participants had excellent adherence to the diet [9]. For children with celiac disease, one research team discovered a strict adherence rate of 44.4%, while another study found that only 40% were poorly adhering to the diet [10,11]. A comprehensive review of the literature found rates of adherence in a range as wide as 23% to 98% for both populations [12]. Although the exact rate of celiac disease patients’ adherence to a gluten-free diet has not been consistently established, each of these compliance rates suggests that a substantial percentage of the celiac disease population cannot consistently meet the treatment guidelines for the disease.

Adhering to a gluten-free diet is strongly associated with a reduction of symptoms for celiac disease patients. A chart review of almost 200 patients found that not adhering to a gluten-free diet was the single strongest predictor of patients with the disease who did not improve [6]. Another study found that children with low dietary adherence significantly complained of more symptoms than those with high dietary adherence [11]. Adherence to the diet can reverse several of the effects of the disease, such as the deterioration in the small intestine, the malabsorption of vitamins, low bone mineral density, and fatigue [6].

Adults with celiac disease represent 0.71% of the United States population [13]. Non-Hispanic Whites, constituting 82.8% of the celiac disease population, are the ethnic group with the highest prevalence rate at 1.01% [13]. Although celiac disease has historically been thought of as a disease primarily experienced by those with European ancestry, its prevalence in other races has increased in recent years [2]. A higher proportion of people in the United States living at or above the latitude of 35° N have celiac disease compared to those living south of this latitude, independent of race or ethnicity [14]. This review did not find documentation of the distribution of celiac disease across rural and urban areas in the United States.

Individuals living in rural areas are less likely to engage in certain health behaviors that help manage chronic conditions. A substantially large study of people living in nonmetropolitan areas of the United States found that those in rural areas engaged less frequently in certain health behaviors associated with the management of the leading chronic disease causes of death [15]. Rural areas also experience higher age-adjusted death rates than metropolitan areas, with a higher number of excess deaths from chronic diseases, including heart disease, stroke, cancer, and chronic lower respiratory illness [16]. This disparity has the potential to directly affect those with celiac disease because the disease is managed nonmedicinally, exclusively through health behaviors [17].

Much of celiac disease dietary adherence research has centered on examining individual factors. While these personal factors can influence dietary adherence, they do not comprehensively address each of the levels of influence that affect health behaviors according to the social-ecological model. The social-ecological model presents a comprehensive theoretical framework addressing the interactions that influence health problems across three levels: (a) the intrapersonal level that includes individuals’ knowledge, attitudes, beliefs, and traits, (b) the interpersonal level that focuses on group interactions that offer identity and support, and (c) the community level that includes institutional, community, and policy factors that influence health behaviors [18].

Several predictive models for celiac dietary adherence have focused on the intrapersonal level by examining demographic information, such as age, annual income, and time since diagnosis, and cognitive elements, such as knowledge, self-efficacy, and risk perception [4,12,19,20,21]. Some researchers have also outlined interpersonal factors affecting adherence, such as membership in a celiac support or advocacy group [11]. However, the existing literature indicates that community-level factors, such as geographic location, can also contribute to poor nutritional adherence.

Hall et al. suggest that intrapersonal factors affect intentional lapses in dietary adherence, while environmental factors affect unintentional lapses in maintaining a gluten-free diet [22]. These environmental factors that act as barriers to adherence can directly affect the lived experience of those with celiac disease, as those who consider a gluten-free diet challenging to follow have a significantly lower quality of life than those who implement the diet with greater ease [23].

A qualitative study showed challenges for rural patients with celiac disease in three areas: (a) optimizing individualized dietician support and services, (b) adapting to a gluten-free diet in a rural context, and (c) managing celiac disease in the context of interpersonal relationships [17]. Inclusion of geographic location with other factors can enhance understanding of identifying those at risk for poor adherence and mean better treatment and support for persons with celiac disease. This study aims to investigate the influence of geographical location on the rural–urban continuum in association with other variables on the dietary adherence needed to manage the celiac disease effectively.

## 2. Materials and Methods

In this study, a combination of a quantitative correlational design and causal-comparative design was used to address the research questions through data collection and regression analysis. We used convenience sampling in a target population of individuals with celiac disease in the United States. Participants were recruited through the National Celiac Association, the Beyond Celiac organization, and an online celiac social media group. After receiving approval from the academic institution’s institutional review board, the researchers began data collection using the online REDCap 13.8.1 survey software where survey data were collected anonymously. Data were only reviewed as demographic variables and a composite CDAT score.

The target population for the study was individuals with celiac disease in the United States. Individuals were excluded from the study if they did not have a medical diagnosis of celiac disease or if they did not have a current address in the United States. Prior to data collection, participants agreed to an informed consent form that specified the purpose of the research, potential benefits and risks, the length of the study, and an option to agree or decline to participate.

### 2.1. Demographic Questionnaire

Geographic location in this study was collected as participant zip codes and coded using the Rural–Urban Commuting Area (RUCA) codes from the U.S. Department of Agriculture [24]. A RUCA code of 1 was designated as urban; a RUCA code greater than 1 was designated as nonurban. The variables of annual income, age, time since diagnosis, and membership in a celiac group were collected through a demographic questionnaire.

### 2.2. Celiac Dietary Adherence Questionnaire

Adherence to a gluten-free diet was evaluated using the Celiac Dietary Adherence Test (CDAT). The instrument addresses five areas of compliance with the gluten-free diet, including disease symptoms, disease knowledge, confidence in the treatment, motivation, and self-reported adherence [25]. The questionnaire has seven items, each rated on a 5-point scale, generating a total score ranging from 7 to 35 points. The scores can be classified into four categories: excellent adherence (7 points), good adherence (8–12 points), inadequate adherence (13–17 points), and poor adherence (over 17 points) [26]. The questionnaire was shown to be highly correlated with the standard dietary evaluation (*p* < 0.001) and outperformed the blood test of immunoglobulin A tissue transglutaminase (IgA) in predicting disease progression [27]. A study found Cronbach’s alpha for the CDAT score was 0.716 with a statistically significant correlation of 0.264 between the CDAT score and IgA (*p* = 0.033) and a strong association between the tool and patient self-reported dietary adherence (*p* < 0.001) [28].

## 3. Results

### 3.1. Sample Characteristics

Responses from 253 participants were included in the final sample. In general, participants lived in an urban area (70.6%), were female (96.5%), and were not a member of a celiac support group (59.2%). Participants had an average age of 48 years, an average income of between USD 60,000 and USD 750,000, and an average of 10 years since diagnosis with celiac disease. The mean CDAT score of the sample was 12.72, with the majority of respondents (57.4%) demonstrating good adherence to a gluten-free diet. The distribution of CDAT scores across the four adherence categories can be seen in Table 1.

Those living in nonurban areas (RUCA 2–10) had a mean CDAT score that represented inadequate adherence (13.06), while those living in urban areas (RUCA 1) had a mean CDAT score that represented good adherence (11.53). The mean age and income level were slightly higher in the nonurban areas, while the mean time since diagnosis was lower than for those living in urban areas, see Table 2.

### 3.2. Group Comparison

An independent sample *t*-test was run on the full sample (*n* = 253) to determine if there were differences in celiac dietary adherence between residents of core urban areas and residents living outside of urban areas. There were no outliers in the data, as assessed by inspection of a boxplot. Dietary adherence scores for each level of geographic location were normally distributed, as assessed by the Shapiro–Wilk’s test (*p* > 0.05), and there was homogeneity of variances, as assessed by Levene’s test for equality of variances (*p* > 0.05). The mean CDAT score for those living outside of urban areas was 13.05 (*SD* = 2.89), representing inadequate adherence to a gluten-free diet. This was higher than the mean score for those living in urban areas of 11.96 (*SD* = 2.88), representing good adherence to a gluten-free diet. This was a statistically significant difference with a moderate effect size for the effect of geographic location on dietary adherence, *M* = −1.083, 95% CI [−1.84, −0.32], *t*(251) = −2.807, *p* = 0.003; *d* = 0.44. Dietary adherence was better for those living in urban areas than for those living outside urban areas.

### 3.3. Regression Analysis

The researchers examined what collection of factors predicts celiac adherence to a gluten-free diet using a multiple regression analysis. For the analysis, the researcher randomly selected 35 participants from each geographic designation using SPSS (some designations contained less than 35 participants, so all were included). The responses in the final sample were grouped into four categories—urban, suburban, small-town, and rural—for inclusion in the regression analysis. Assumptions were assessed prior to conducting the regression analysis to ensure the data were appropriate for the planned analysis. Based on regression diagnostics, we determined the multiple regression model was a good fit for the data. The *R* for the overall model was 24.1% with an adjusted *R*^2^ of 19.1%, indicating that the model explained nearly 20% of the overall variance for dietary adherence. Age, time since diagnosis, income, group membership, and geographic location statistically significantly predicted dietary adherence, *F*(7, 105) = 4.774, *p* < 0.001.

## 4. Discussion

### 4.1. Discussion of Results

This study aimed to investigate the influence of rural–urban geographic location of residence in association with other variables on the dietary adherence needed to manage celiac disease effectively. In this study, there was a statistically significant difference with a moderate effect size between the dietary adherence of those in urban and nonurban areas. People living in urban areas had a mean CDAT score in the category of good adherence to a gluten-free diet. Those living in nonurban areas had a mean CDAT score representing inadequate adherence to a gluten-free diet. Thus, those with celiac disease living in nonurban areas might potentially have consumed, on average, enough gluten to risk disease progression. In contrast, those in urban areas on average appear to comply with the diet adequately enough to avoid disease progression and further complications. This significant difference in dietary adherence could result from reduced access to gluten-free foods, health care, and interventions outside of metropolitan areas.

This finding is in contrast to the study by Zanini et al., who found no difference in dietary adherence by geographic area between urban and extra-urban areas [29]. However, these results support the qualitative findings of Lee et al., who identified barriers to dietary adherence that exist in nonmetropolitan areas, as well as the findings of Matthews et al., who found that those outside urban areas engage less frequently in health behaviors associated with the management of chronic diseases [15,17].

The rate of adherence to the gluten-free diet among celiac patients varies highly in the literature, with a comprehensive review finding adherence rates ranging from 23% to 98% [12]. In this study, 59.3% of the sample in this study overall had good or excellent adherence to a gluten-free diet, at a rate similar to Halmos et al., who found that 61% of the participants in their sample were adherent to a gluten-free diet [4]. The current study found that only 1.2% of the sample had excellent adherence, lower than the 6% excellent adherence found by Gładyś et al. using the same instrument [26].

Data analysis for factors affecting dietary adherence revealed that the percentage of variance accounted for by the regression model was 19.1% when all independent variables were included in the model. This analysis included a breakdown of geographic location across four categories—urban, suburban, small-town, and rural. The overall effect size was moderately high. When the regression analysis was conducted with the geographic location of participants analyzed across only two categories, urban and nonurban, the effect size was reduced by almost half, suggesting the need to have finer categories of geographical location for precision.

Household income added statistically significantly to the regression equation as a predictor of dietary adherence. Higher annual incomes predicted better adherence to the diet. This finding is supported by observations from White et al., showing that the cost of gluten-free foods is one of the primary barriers to dietary adherence [30]. This also aligns with the correlation found Muhammad et al. and Villafuerte-Galvez et al. between the perceived cost of the gluten-free diet and dietary compliance [31,32]. Those with a lower income may struggle to afford gluten-free foods, affecting their ability to maintain a gluten-free diet.

This study also found that age was a significant predictor for dietary adherence, with older participants demonstrating better adherence to the dietary guidelines. This is similar to the findings by Fueyo-Díaz et al. and Halmos et al., who found that older age in adults resulted in better adherence [4,20]. A longer time since diagnosis also contributed to better dietary adherence, which agrees with the finding by Fueyo-Díaz et al. [21]. While it could be posited that increased compliance with age results from a longer time since diagnosis, this study did not find a correlation between the age of participants and the length of time since diagnosis.

Those living in urban areas had a longer average time since diagnosis than those in nonurban areas by about 4 years, while the average age was similar. Celiac disease is a condition that often takes considerable time to be diagnosed; therefore, a more extended time since diagnosis for those living in urban areas may indicate that they have greater access to medical resources, resulting in earlier diagnoses.

### 4.2. Limitations

The study relied on responses from volunteers, with the recruitment materials distributed to individuals in celiac organizations or social media groups. This could have introduced selection bias in that the potential participants who are actively engaging with the materials from advocacy groups or disease-specific social media forums may respond differently to the study instrument items than those across the general celiac community as a whole. Beyond this, an increased sample size with higher participation rates from nonurban populations and male participants could have resulted in a more representative sample.

The location of participants was a limitation in this study because the data collection was completed across a variety of settings, which could have influenced the participants’ responses. Because it is impossible to include every known variable in the regression model, this researcher could have overlooked essential variables affecting the results. While this researcher attempted to include a set of variables with solid support in the existing literature across the domains of the theoretical framework, the model only accounted for about 20% of the variance, meaning additional significant variables contributing to the outcomes could have been included.

The geographic location data were collected at the zip code level and coded on a continuum of urban-to-rural based on census data from 2010 [24]. However, not all zip code areas are homogenous; therefore, a more granular level of analysis could produce more accurate results. Additionally, the population in certain regions may have shifted since the codes established from the 2010 data set. While census data from 2020 had been collected at the time this research was conducted, the rural-to-urban classification by zip code had not yet been completed and was not available to be used in this study.

Membership in a celiac group was not a significant contributor to the regression equation and was excluded from the forward regression equation. This was in contrast to multiple studies that found such group membership to affect dietary adherence significantly [25,32,33,34]. The study survey did not define support groups or specify digital compared to in-person groups, which could lead to a varied response and lack of significance. The demographic questionnaire could have been improved by asking if the participants were also members of a celiac association.

### 4.3. Recommendations for Future Research

The study findings can be used to support the efforts of public health and healthcare practitioners to focus efforts for intervention at a population and individual level, respectively. The concentration of efforts and resources in nonurban areas has the potential to further increase gluten-free dietary adherence and positively affect celiac disease symptoms through policies, initiatives, and programs that encourage diversity in the food supply, better surveillance, educational promotion, and treatment. While a focus on nonurban areas is indicated, continued efforts to promote gluten-free diets in urban areas, especially in younger patients and those with lower household incomes, may also be warranted.

Based on this study’s findings, researchers developing interventions to improve dietary adherence for those with celiac disease should give special attention to the areas of most need: those outside of an urban region, younger patients, those with lower household incomes, and those more newly diagnosed. This supports the need for interventions that can be applied across nonmetropolitan areas at a lower cost and are more relevant to the needs of younger generations, such as digitally based educational resources.

For future research, clinical studies with measures of levels of tTG or intestinal biopsies can be used to monitor program outcomes. The study could be expanded to include additional variables beyond those in this regression analysis in accordance with the geographic location variable to create models that account for even more of the variation in celiac dietary adherence. Additionally, we examined the effects of dietary adherence and geographic location within the context of celiac disease. Future researchers could examine geographic location within the context of other health conditions with specific dietary adherence requirements, such as diabetes, heart disease, Crohn’s disease, or food allergies.

## 5. Conclusions

Those living in urban areas had, on average, good adherence to a gluten-free diet, while those living in nonurban areas had, on average, poor adherence to the diet. Those in nonurban areas face an increased risk for ongoing physical damage and disease progression due to the dietary adherence rates found. The study findings provide support to public health and healthcare practitioners in their efforts to intervene at both the community and individual levels to support those with celiac disease. By directing their efforts and allocating resources to nonurban areas, among younger patients and those with lower incomes, there is potential to enhance gluten-free dietary adherence and positively impact symptoms of celiac disease.

## Figures and Tables

**Table 1 nutrients-15-04535-t001:** Dietary adherence of sample.

CDAT Adherence Category	*n*	%
excellent adherence	3	1.2%
good adherence	147	58.1%
inadequate adherence	88	34.8%
poor adherence	15	5.9%

**Table 2 nutrients-15-04535-t002:** Means and standard deviation of persons with celiac disease by area of residence.

Variable	Urban *M* (*SD*)	Nonurban *M* (*SD*)
CDAT score	11.53 (3.1)	13.06 (2.9)
age (years)	47.4 (15.5)	48.8 (14.6)
annual income level (bracket)	12.26 (4.9)	12.35 (4.3)
time since diagnosis (years)	12.1 (10)	8.9 (8.4)

## Data Availability

The data presented in this study are available on request from the corresponding author.

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
