# Peer review of "Celiac Disease Dietary Adherence on the Rural–Urban Continuum"

_nutrients, 2023, doi:10.3390/nu15214535_

Round 1

Reviewer 1 Report

Adherence to the strict gluten-free diet can be challenging and adherence rates are insufficient. Therefore, the subject of the manuscript is of importance.

However, some issues in the manuscript need to be considered and improved. 

Title: I suggest including adherence in the title as this is a main aspect of the study aim. 

Introduction: In general, the introduction should be more thorough and cover more depth of the literature. 
The description of the three levels from line 43 could be presented more clearly. For example, (a) the intrapersonal level that is comprised of....
Materials and Methods: "After receiving exempt status...." Why was except status given? was the study anonymous? This sentence is unclear when reading further on in the manuscript. 

I suggest considering presenting the demographic questionnaire first and then the CDAT. Were there additional demographic questions in the questionnaire? Also I suggest adding a subtitle before each questionnaire.
Results: Line 97 - "...and not a member of a celiac support group". But were they members of the celiac associations? 
Line 98 - add "years" after 48. 

Line 105 - "Those living in nonmetropolitan areas". How is this defined in the USA? For example, less than how many residents? According to what criteria? 

Line 109 - "see Table 2". It would be clearer if these results were presented as a comparison urban vs nonurban in a more organized way.
Discussion: In general the discussion section is lacking connection to relevant previous research.  
Line 166-167 - This important information re adherence is more suited to the introduction section. 
Line 207: Add subtitle "limitations"
Institutional Reveiw Board Statement: This is unclear. Was the study approved or exempted?
Data Availability Statement: "The data are not publicly available due to the terms of informed consent and IRB review recommendations". Again this is unclear. If exempt from need for approval, how can there be terms of confidentiality? 

Reviewer 2 Report

This study aims at describing differences in dietary adherence to the gluten-free diet (GFD) between populations living in urban vs. non-urban locations. This reviewer has the following comments/criticisms:

Major

I. As the authors do admit, the sample size is quite limited to reach definitive conclusions.  A larger study with the same methodology would have been more credible.

II. The conclusions that "This means those with celiac disease living in nonurban areas consumed, on average, enough gluten to cause ongoing physical damage and disease progression. In contrast, those in urban areas were able to comply with the diet adequately enough to avoid disease progression and further complications." appear excessive and not based on evidence. Please rephrase to more reasonable terms, something like "Thus, those with celiac disease living in nonurban areas might potentially have consumed, on average, enough gluten to risk disease progression. In contrast, those in urban areas on the average appear to comply with the diet adequately enough to avoid disease progression and further complications."

III. While I am no statistician, I find it hard to believe that there is a statistically significant difference ("p=0.03") between 13.05 (with a large SD of 2.89!!) and 11.96 (with an equally large SD of 2.88). Would the authors please comment? 

Minor

I. Table 3 is rally too technical to be useful to the readers. I respectfully suggest deleting it.

II. The sentence "This finding is supported by the observations from that the cost of gluten-free foods is one of the primary barriers to dietary adherence and the correlation between the perceived cost of the gluten-free diet and dietary compliance" is cumbersome and awkward. Please rephrase to make it clear.

Round 2

Reviewer 1 Report

The authors sufficiently addressed all comments.

Author Response

Thank you for taking the time to provide comments and feedback. 

Reviewer 2 Report

The revised version addresses adequately most of my remarks. I am however still of the opinion that Table 3 adds undue weight to the readability of the paper. 

Author Response

Thank you for your comments. Table 3 has been removed from the manuscript as suggested.